# Stress Responses to One-Day Athletic Tournament in Sport Coaches: A Pilot Study

**DOI:** 10.3390/biology11060828

**Published:** 2022-05-27

**Authors:** Zbigniew Obmiński, Jan Supiński, Łukasz Rydzik, Wojciech J. Cynarski, Mariusz Ozimek, Zbigniew Borysiuk, Wiesław Błach, Tadeusz Ambroży

**Affiliations:** 1Department of Endocrinology, Institute of Sport—National Research Institute, 01-982 Warsaw, Poland; zbigniew.obminski@insp.waw.pl; 2Faculty of Physical Education & Sport, University School of Physical Education, 51-612 Wroclaw, Poland; jan.supinski@awf.wroc.pl (J.S.); wieslaw.blach@awf.wroc.pl (W.B.); 3Institute of Sports Sciences, University of Physical Education, 31-571 Cracow, Poland; mariusz.ozimek@awf.krakow.pl (M.O.); tadek@ambrozy.pl (T.A.); 4Institute of Physical Culture Studies, College of Medical Sciences, University of Rzeszow, 35-959 Rzeszów, Poland; ela_cyn@wp.pl; 5Faculty of Physical Education and Physiotherapy, Opole University of Technology, 45-040 Opole, Poland; z.borysiuk@po.edu.pl

**Keywords:** judo, tournament, coaches, stress, hormones, cardiovascular responses

## Abstract

**Simple Summary:**

Judo coaches respond to stress during one-day tournaments, as evidenced by significant changes in hormonal status. The cumulative effect of stressful stimuli was noticeable especially at the end of the day. The results of the present study indicate the usefulness of using tools and a way to study stress among coaches. A similar research protocol can be used to assess stress responses to single-day stimuli associated with other social and occupational roles in healthy middle-aged men.

**Abstract:**

Background: Watching athletic tournaments induces emotional and physiological responses in sports fans and coaches. The aim of the study was to investigate hormonal and cardiovascular responses in judo coaches observing the fights of their athletes during one-day, high-level tournaments. Material and methods: We studied the biological responses of a group of seventeen judo club coaches between the ages of 39 and 57 years to a one-day judo tournament attended by the adult male and/or female athletes that they coached. At the time of the tournaments and also in neutral conditions, the coaches’ capillary blood and saliva samples were collected concurrently two times a day, both at 7:30 a.m. and after completion of the tournament (at about 8:00 p.m.). Cardiovascular parameters were also determined at the same times of day. Sleep quality (SQ) was assessed on a 6-point scale both on the night preceding the tournament and in neutral conditions. Cortisol (C) and testosterone (T) levels were determined from serum and saliva samples. The results for both days at different times of day were compared. Results: Statistical calculations showed higher concentrations of cortisol and a greater reduction of testosterone levels in serum and saliva at the end of the day during the tournament compared with those on a neutral day. Morning and evening cardiovascular indices were higher during the tournament. Conclusion: The study showed that during one-day tournaments, judo coaches were exposed to stress that induced intermittent hormonal changes in blood and saliva and activated the autonomic nervous system.

## 1. Introduction

High-level sporting events watched directly or in the media arouse strong emotions among viewers, which in some persons can increase the risk of cardiac incidents. Research findings on the risk of myocardial infarction in football fans during matches are inconsistent [1,2,3,4,5,6]. The risk of serious cardiac disorders in healthy middle-aged men exposed to psychological stress depends on the type of their personality. Coronary heart disease or myocardial infarctions occur more frequently in people who tend to experience strong and negative emotions (anger, fear, sadness, and frustration) in response to stressful mental stimuli [7,8,9,10].

The effect of emotions on physiological changes has been evaluated in studies on the intensity of emotions. It has long been known that temporarily experienced psychological stress induces hormonal and cardiovascular responses, while long-lasting psychological stress may be responsible for the development of chronic hypertension. This phenomenon has been used in laboratory settings to test the emotional and physiological responsiveness to psychosocial stress stimuli. The most commonly used standardized stimulus is a test known as the Trier Social Stress Test (TSST), which examines the responses of the cardiovascular system such as heart rate (HR), systolic blood pressure (SBP), diastolic blood pressure (DBP), and changes in salivary cortisol levels. After experiencing stimuli, the above-mentioned parameters increase to an extent dependent on the type of personality [11,12,13,14,15] or level of physical fitness [16,17,18,19].

Sporting events are strong stimuli, not only for the fans but also for coaches watching their athletes. The tests conducted on football coaches during matches showed changes in salivary cortisol levels and IgA [16] or alpha-amylase [17]. In swimming coaches, watching tournaments caused changes in blood fibrinogen levels [18]. It seems that tournaments at higher levels induce major responses in viewers. The higher the level of the tournament, the greater the physiological responses are. During the 2010 World Cup, viewers of the final soccer match demonstrated significantly higher cortisol and testosterone levels compared with cortisol and testosterone levels under neutral conditions [19]. Rumination during the night preceding athletes’ participation in the tournament decreases sleep quality in athletes [20,21], similar to workers experiencing occupational stress [22]. A stronger opponent causes a greater mental and hormonal response before the beginning of a sporting event [23]. Parallel research on coaches and their athletes taking part in tournaments showed the interaction of emotions in both groups. It manifests itself as a positive correlation on the level of anxiety before a tournament in athletes and their coaches [24]. The strength of this relationship depends on the social style and relations between the coach and the athlete [25,26], and it may have crucial consequences for sports performances [27].

Soccer games are relatively short compared to one-day-long judo tournaments. In the available literature, there is no data on how longer watching sporting events induces biological responses in coaches. One-day-long judo tournaments are an example of such a situation. Although a single official judo bout takes a few minutes on average, an athlete can participate in several bouts during the day (finalists may participate in up to six bouts). Furthermore, in athletes and their coaches, pre-fight emotions may appear in the evening on the day before the athlete’s participation in the event when the list of opponents in the first fight is presented. Too-early anticipation of the first fight may appear in judokas on the night before the tournament, especially when the opponent is highly ranked on the world’s ranking list. On the day of the tournament, emotions in judo athletes and coaches are intense in the morning, during the weigh-in, and then from the start of the first elimination fights (approx. 10:00–11:30 a.m.) until the evening, when the final fights take place. A two-hour break in the tournament separates the first and second rounds of elimination fights. Similar to the findings of previous authors, one might expect that a one-day judo tournament is a mental stimulus that is strong enough to elicit noticeable physiological responses in coaches. Repeated experiences of this kind have some health consequences. Furthermore, the overall psychological well-being of coaches is influenced by their athlete’s performance. Athletes’ failure and coaches’ sense of high responsibility and overwork are among the common reasons for the burnout syndrome in this professional group [28,29].

The aim of the study was to investigate biological responses to stress in coaches caused by watching judo athletes fighting during a one-day high-level tournament.

## 2. Materials and Methods

### 2.1. Participants

We investigated a group of 17 Polish judo club coaches aged 39–57 years with a body height of 169–183 cm and body mass of 74–88 kg, who were healthy, physically active, and non-smokers. They accompanied their athletes during a one-day international judo tournament in Warsaw. Athletes and coaches stayed at the tournament venue in the evening on the day before the event. At that time, the organizer of the tournament provided the list of all competitors, including the names of opponents paired and competing in the first bout of the tournament. According to the rules, the first defeat excluded an athlete from subsequent competition.

### 2.2. Measurement Methods

Using automatic sterile lancets and under aseptic conditions, capillary blood samples from the earlobe were collected during the tournament two times, once in the morning (approx. 07:30 a.m.) and once in the evening after the closing ceremony (8:00 p.m.). Concurrently, at the same time points, saliva samples were collected by passive drooling into plastic tubes. Next, heart rate (HR), diastolic blood pressure (DBP), and systolic blood pressure (SBP) were measured in the sitting position using an electronic device. Prior to saliva sampling, the subjects rinsed their mouths with distilled water. Biological material was collected before meals (breakfast, lunch, and dinner). The same procedure was applied to coaches in the controlled conditions considered neutral, i.e., at the time of routine activities in clubs or training camps.

### 2.3. Experimental Design

All the collected biological material was frozen until cortisol and testosterone assay. Assays were performed using commercial kits for the determination of hormone concentrations based on the ELISA method using DRG-GERMANY kits. Before the analysis of hormones in saliva, the samples were thawed and centrifuged (3000 rpm) for 5 min to obtain a homogeneous solution with reduced viscosity. Determinations of serum cortisol (Cser) and testosterone (Tser), as well as salivary cortisol (Csal) and testosterone (Tsal) were performed in duplicates in one run to avoid the effect of between-assay variation. Based on pairs of single measurements, analytical within-assay variation, also known as within-run imprecision and expressed as CV%, was 8.9% for Csal, 9.5% for Tsal, 5.1% for Cser, and 6.3% for Tser. On the night before the tournament and at the time of control tests, coaches evaluated the quality of their sleep on a 6-point scale (from 1 point being insomnia to 6 points being deep and continuous sleep). Sleeping time was recorded from going to bed to awakening. On the day of the tournament, the awakening of all the coaches was not spontaneous but was triggered by an alarm clock. The relationship between coaches’ age and resting testosterone levels in blood collected on the morning of the control day is presented as a linear regression equation.

### 2.4. Bioethics Committee

Prior to participation in the tests, the coaches were informed about the research procedures, which were in accordance with the ethical principles of the Declaration of Helsinki WMADH (2013). Obtaining the competitors’ written consent was the precondition for their participation in the project. The research was approved by the Bioethics Committee at the Regional Medical Chamber (No. 287/KBL/OIL/2020).

### 2.5. Statistical Analysis

A two-way analysis of variance (daytime × situation) was used to compare the differences between the data at different times of the day and in two situations. The difference in the quality of sleep in both cases was assessed using the Wilcoxon signed-rank test. Statistical calculations were made on log-transformed data using STATISTICA software (Kraków, Poland), version 13.1. The level of significance was set at *p* < 0.05. The Shapiro–Wilk test confirmed a normal distribution of hormonal variables. The relationship between participants’ age and testosterone levels in the blood collected on the morning of the control day is presented as linear regression.

## 3. Results

The C and T levels, values of anabolic–catabolic indices (T/C × 10^2^) in saliva and serum in the morning and evening, cardiovascular status on the control day and during the tournament, and results of statistical calculations are shown in Table 1. Table 2 presents cardiovascular parameters (HR, SBP, and DBP) in a similar manner. The relationship between hormone levels in serum and saliva for cortisol and testosterone are given as regression equations. The *p*-value for significant differences between means is highlighted in bold. The exact *p*-value is given for non-significant differences (ns).

The average quality of sleep on the night before the tournament (4.6 ± 0.5) was significantly worse (Z = 3.05, *p* < 0.002) than the average quality measured in neutral conditions (5.5 ± 0.5). Mean nocturnal sleep time under neutral conditions (7.6 ± 0.6 h) was significantly longer (Z = 3.62, *p* < 0.000) than the mean nocturnal sleep time the night before the tournament. It should be noted that the coaches who used to take naps (up to 1 h) after lunches during the training camps or at home did not have such an opportunity at the competitions, which caused psychological discomfort. Each of the 17 coaches assisted one male or female athlete who was involved in a different number of bouts (1–5). Eleven coaches watched more than one of their athlete’s fights (2–5 fights). In total, this group watched 33 fights, including 22 wins and 11 losses. Each fight lost by a male or female athlete ended their participation in the tournament. Judokas trained by the other four coaches lost their first fights and were excluded from subsequent bouts. The athletes of the next two coaches, despite losing their second bouts, were given the right to the next fight through repechage. All the coaches also observed the fights of other athletes, including all final fights taking place at the time of the second round of the tournament. When specifying the types of emotions when observing fights of their athletes, the coaches studied stated that directly before these fights, the predominant emotions were their concern about the result and their anger following a defeat.

The average hormone levels decreased during the day, both at the time of the tournament and on the control day. It is worth noting that these changes occurred in accordance with the well-known diurnal rhythms. In the evening, the serum cortisol levels on the control day were 54% and 70% of the morning values on the control day and on the day of the tournament, respectively. A similar pattern of changes was recorded for cortisol levels in saliva. The daily average cortisol level was significantly higher on the day of the tournament than on the control day. The relative decrease in testosterone levels in serum and saliva during the day was less pronounced and not statistically significant. The relative differences between concentrations of testosterone in serum in the morning and the evening on the control day and the day of the tournament were 22% and 35%, respectively.

Two models describe the relationship between cortisol in saliva and serum:

Model 1 is based on a second-order polynomial: C saliva = 9.09 − 0.0040 × C serum + 0.00006 × (C serum)2, with a determination coefficient of R2 = 0.786.

Model 2 consists of two linear regressions which depend on ranges of serum cortisol, C serum < 400 nM, C saliva = 7.389 + 0.0189 ×, C serum, R2 =0.656, *p* < 0.000, C serum > 400 nM: Csaliva = 3.829 + 0.0374 × C serum R2 = 0.291, *p* < 0.003.

The linear relation of saliva–serum testosterone is given by the equation:T saliva = 0.0046 + 0.0094 × T serum, at R2 = 0.258, *p* < 0.008

The relationship between age and morning testosterone levels is expressed by a linear regression: Tserum = 26.166 − 0.1305, at R = 0.115, *p* = 0.077 for intercept and 0.660 for regression slope, respectively.

## 4. Discussion

Cortisol and testosterone have been frequently studied under a variety of circumstances, in both neutral conditions and stress. Analysis of the results must take into account the fact that the levels of these hormones depend on many factors that can mask the studied effect of psychophysical stimuli. In healthy middle-aged men (average age of 47.2 years), no relationship has been reported between hormone levels and BMI, and some effect of age was revealed only for a very wide range of 20 to 80 years for this variable [30].

Hormonal activity on the control day in judo coaches was consistent with the well-known dynamics of changes in cortisol levels [31]. Similarly, blood testosterone levels followed a diurnal rhythm observed in healthy men in neutral conditions [32]. The highest concentrations of both hormones were observed in the morning, and the hormone levels were decreased until late in the evening. This physiological phenomenon is defined as the diurnal rhythm of hormone levels. However, exposure to strong stimuli may distort the diurnal rhythm of cortisol levels [31].

In judo competitions, the highest psychological stress in a competitor is usually induced by waiting for his or her first fight, since the result of this fight determines the subsequent participation of an athlete in the tournament. Likewise, the coach experiences the same emotional state, a mix of anxiety and/or anger, depending on the athlete’s behavior. For that reason, it is impossible to perform any observation at this moment that might disturb the athletes’ and their coaches’ attention directly before the fight.

The activity of both hormones was already consistent with their diurnal rhythm, but a quantitative difference in the dynamics of these changes was found. On the day of the tournament, cortisol levels were decreasing more slowly, whereas testosterone levels were decreasing faster compared with neutral conditions. This difference suggested that coaches were still exposed to stress associated with the tournament.

Particular attention should be paid to the interpretation of changes in testosterone levels during the day of the tournament. Several observations suggest that short-term psychosocial stress stimulates the release of this hormone into the blood. This has been shown in studies on football fans, who had elevated levels of the hormone from the beginning to the end of the 2 h match [19]. On the other hand, a higher decrease in the concentration of androgen throughout the day during judo tournaments indicated that a longer exposure to stress partially inhibits the secretion of testosterone. This phenomenon consists of the inhibiting effect of elevated cortisol levels on luteinizing hormone (LH), which stimulates testosterone biosynthesis in men. This phenomenon was observed in men, who, throughout the night, were exposed to anticipatory stress [32]. Higher cortisol and lower testosterone levels during the tournament led to a decrease in the anabolic–catabolic index on that day. This meant that the emotional stress induced by the tournament negatively affected the metabolic balance, thus exacerbating the protein catabolism processes, particularly at the end of the day.

The curvilinear relationship between salivary and serum cortisol levels was similar to those shown in previous studies [33,34,35]. Some studies have found a relationship between salivary and serum cortisol as a function with an inflection point, which belongs to the two linear regressions, one with a low slope and one with high slope. This inflection represents a level of total serum cortisol when serum corticosteroid-binding globulin (CBG) is fully saturated by the ligand [35]. As shown in our study, the symptoms of CBG saturation were observed at serum cortisol levels near 400 nM. The salivary levels were many times lower than those in serum, but there is a belief that they reflect free biologically active fractions of hormones. Consequently, changes in salivary cortisol provide information about glucocorticoid activity in blood, whereas changes in total serum cortisol are a symptom of the current activity of the adrenal cortex, i.e., the rate of hormone secretion from the zona fasciculata. Hence, both assays in saliva and serum are of diagnostic value in the estimation of stress responses to a challenge.

Cardiovascular responses in coaches to tournament fights of their athletes, such as elevated SBP, DBP, and HR, were qualitatively and quantitatively similar to the responses to the laboratory-standardized stress [11,12,36]. The anticipation of tournaments significantly reduced the quality of sleep for coaches, and this could have caused the slightly increased blood pressure in the morning. Sleep difficulties due to daily stress at work (occupational stress) are fairly well described in the literature [22,37]. In such cases, poor quality of sleep often precedes chronic fatigue syndrome. The presence of coaches during high-level tournaments is not common; therefore, incidental stress-inducing situations do not pose a risk to health.

In our research, we found significant individual differences in biological responses throughout the tournament. This may be due to different emotional reactivity, type of personality, and degree of involvement of coaches, especially the “coach–athlete” relationships, which could facilitate the interpretation of individual results [38]. Another source of this between-subject variation might be various physical fitness among the coaches studied. It is known that better fitness and higher leisure-time physical activity are responsible for reduced susceptibility to psychosocial stress [39,40,41]. The type of personality is also a factor in various stress responses to stimuli.

Progressive changes observed with age in androgenic status in blood in men are referred to as andropause and defined as androgen deficiency in the aging males. Consequently, 20% of men aged >60 years and 50% of men aged >80 years have testosterone levels below the reference value set for young adults [42]. Many authors have reported only a slight decrease (1% per year) in the level of this androgen with age for a wide range (from 20 to 80 years) [43,44,45]. Above the age of 40 years, the decrease in the androgen levels is imperceptible, but the variability of this parameter increases [46], which may be related to the deteriorating health status of aging men. In our study, we analyzed the linear relationship between age and testosterone levels. We did not find a significant decrease in its level in coaches. It appears that the range of the independent variable (age) was too narrow in our study, the sample size too small, and the health status in all participants too similar for an age–testosterone relationship to be observable. There were also no age-dependent relationships between hormonal and cardiovascular responses.

Anticipation of competition significantly adversely affected nocturnal sleep for coaches. Sleep quality deteriorated, and sleep time was shorter. This showed that an important event such as a tournament affects the sleep characteristics at night not only in athletes, which has been shown in various sports in athletes [47], but also in individuals involved in the preparation for the competition and responsible for its outcome. It is also likely that the quality of the night’s rest is responsible for the elevated morning blood pressure and HR on the day of a tournament.

Study limitation:The study lacks information about the personality profile of the participants. Stable personality traits can modulate physiological responses to stressful stimuli, as reported in the cited literature [11,12,13,14,15]. With such a small sample, it would be difficult to establish a clear relationship between personality and hormonal and cardiovascular responses. Furthermore, the stimuli were highly variable, e.g., depending on the number of bouts observed (two to five) and their total duration.The coaches declared considerable physical activity, but a reliable assessment of such behaviors requires the use of a special questionnaire.After completion of the tournament, it was impossible to examine the overall mood in coaches, which would be expected to reflect an assessment of the extent to which the tournament was considered successful.The participants declared their basic anthropometric characteristics such as body height and body weight, but these data were not verified by our measurements. In addition, body components such as the percentage of fat tissue, muscle mass, and bone mass were not measured.Our study compared the same specific study group in two different situations, i.e., exposed to stress and in a neutral situation. Therefore, it is difficult to generalize study results to responses to stress expected in other occupational groups. Furthermore, with the small sample size, the study can be qualified as a pilot study.The study did not assess levels of post-competition satisfaction/frustration.

## 5. Conclusions

Judo coaches present during one-day-long tournaments in which their athletes participate responded to stress with a significant change in the hormonal status and cardiovascular system.The cumulative effects of stress-inducing stimuli related to the tournament were visible, especially at the end of the day, and were expressed by a significant hormonal imbalance, i.e., disruption of physiological rhythms of cortisol and testosterone levels.The results show the usefulness of the applied tools and the research mode of stress in judo coaches. A similar research protocol can be used to assess the stress responses to one-day-long stimuli connected with the performance of other social and professional roles in healthy middle-aged men.

## Figures and Tables

**Table 1 biology-11-00828-t001:** Descriptive statistics for hormones, anabolic–catabolic values in saliva and serum, and differences between days (control day vs. competition) and time of day (morning vs. evening).

Variable	Condition	Morning	Evening	*p*-Value	Δ%
C serum (nM/L)	Control day	426 ± 76	232 ± 66	<0.001	45.5
	Competition	487 ± 70	342 ± 60	<0.000	29.8
	*p*-value	0.237 ns	<0.001	-	-
	Δ%	−14.3	−4.4	-	-
C saliva (nM/L)	Control day	19.1 ± 4.3	12.0 ± 3.3	<0.000	37.2
	Competition	21.7 ± 5.2	14.1 ± 2.9	<0.000	35.0
	*p*-value	0.284 ns	0.109 ns	-	-
	Δ%	−13.6	−17.5	-	-
T serum (nM/L)	Control day	21.0 ± 5.6	16.4 ± 3.8	<0.05	21.9
	Competition	20.2 ± 4.4	13.0 ± 2.2	<0.005	35.6
	*p*-value	0.988 ns	<0.000	-	-
	Δ%	3.8	20.7	-	-
T saliva (nM/L)	Control day	0.257 ± 0.079	0.196 ± 0.066	<0.05	23.7
	Competition	0.244 ± 0.100	0.144 ± 0.062	<0.01	40.0
	*p*-value	0.992 ns	<0.003	-	
	Δ%	5.1	26.5	-	
T/C × 10^2^ serum	Control day	5.00 ± 1.42	7.62 ± 3.47	<0.05	−52.0
	Competition	4.20 ± 0.97	3.92 ± 1.08	0.911	7.1
	*p*-value	0.988 ns	<0.000	-	
	Δ%	16.0	47.2	-	
T/C × 10^2^ saliva	Control day	1.36 ± 0.63	1.60 ± 0.57	0.517	−17.6
	Competition	1.28 ± 0.77	1.02 ± 0.38	0.646	20.3
	*p*-value	0.984 ns	0.002	-	-
	Δ%	5.9	37.5	-	-

C: cortisol; T: testosterone; T/C: changes in the T/C ratio.

**Table 2 biology-11-00828-t002:** Descriptive statistics of heart rate (HR), systolic blood pressure (SBP), and diastolic blood pressure (DBP) and the respective differences between means.

Variable	Conditions	Morning	Evening	*p*-Value	Δ%
HR	Control day	65.0 ± 6.0	66.8 ± 4.2	0.711	2.8
	Competition	72.9 ± 5.6	69.9 ± 6.3	<0.05	4.1
	*p*-value	<0.05	<0.05	-	
	Δ%	−12.2	−3.1	-	
SBP	Control day	122.3 ± 6.8	125.1 ± 7.3	0.105	−2.5
	Competition	133.7 ± 11.4	137.8 ± 10.6	<0.05	3.0
	*p*-value	<0.01	<0.01	-	
	Δ%	−9.3	−10.2	-	
DBP	Control day	66.6 ± 6.7	67.2 ± 6.5	0.993	0.09
	Competition	71.5 ± 6.3	73.5 ± 7.6	0.496	2.7
	*p*-value	<0.05	<0.05	-	
	Δ%	7.4	6.3	-	

HR: heart rate; SBP: systolic blood pressure; DBP: diastolic blood pressure.

## Data Availability

All data are presented in the study.

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
