# Peer review of "Stress Responses to One-Day Athletic Tournament in Sport Coaches: A Pilot Study"

_biology, 2022, doi:10.3390/biology11060828_

Round 1

Reviewer 1 Report

Dear Authors

I write you in regards to manuscript # biology-1685661 entitled "Stress Responses To One-Day Athletic Tournament In Sport Coaches. A Pilotage Study" which you submitted to the biology. This paper is considered to be more suitable for the field of 'physiology' than 'biology'. Anyway, I would like to inform you of the consideration of your manuscript as follows.

Minor points

In all tables, there are many things that should be entered as a 'decimal point' with a 'comma'.

Major points

1. This paper compares the pre- and post-level levels for one group without a control group, which causes a serious problem in generalizing the results of this paper.

2. Normality test and sample count calculation are omitted in this paper.

Sincerely,

Author Response

Dear Reviewer,

Thank you very much for your time and valuable comments, which all have been considered and incorporated. The detailed list of responses is given below. We hope that the modifications and explanation will be acceptable for you.

Yours sincerely,

Rydzik, corresponding author

Dear Authors

I write you in regards to manuscript # biology-1685661 entitled "Stress Responses To One-Day Athletic Tournament In Sport Coaches. A Pilotage Study" which you submitted to the biology. This paper is considered to be more suitable for the field of 'physiology' than 'biology'. Anyway, I would like to inform you of the consideration of your manuscript as follows.

A: Dear reviewer, thank you for your review. We would like to point out that the journal is called Biology but has a much broader scope. The section to which we sent the article is called "physiology".

Minor points

In all tables, there are many things that should be entered as a 'decimal point' with a 'comma'.

A: This has been corrected

Major points

  1. This paper compares the pre- and post-level levels for one group without a control group, which causes a serious problem in generalizing the results of this paper.

A: This study is not a typical experiment but an observation. Therefore, the study group is itself a control group.

  1. Normality test and sample count calculation are omitted in this paper.

A: This has been corrected

Reviewer 2 Report

As far as general concepts are concerned, in my opinion, this article is carried out using a correct method and contributes to the knowledge of how these parameters can vary in coaches who observe their competitors during a full day in high-level tournaments where they must overcome different eliminatory rounds. In addition, knowing the results found in the cardiovascular parameters can be very important information for those coaches at risk of suffering cardiovascular accidents.

More specifically, I think the following errors should be corrected:

  • In Table 1 there are some results with a level of significance p >0.05 that are accompanied by (ns) and others that are not. The meaning of "ns" does not appear in any legend of the table, so I deduce that its meaning is "not significant", but I think that it must be present in all the results with p > 0.05 or in none of them, since it can be deduced that p>0.05 is a non-significant result, as explained in the statistical analysis section (line 126). Similarly, as mentioned above, I think that if (ns) is included for non-significant results, it should appear in a table legend.
  • On the other hand, in Table 1 the results of significance with p <0.05 are highlighted in bold, but, however, in Table 2 all the results of significance are highlighted in bold, both those with p <0, 05, like those of p>0.05.
  • Finally, in line 91 it is explained that 17 coaches were investigated and yet in line 139 it states that 11 coaches observed more than one fight of their competitors and that the judokas of 4 coaches lost in their first fight and were excluded of the tournament, which adds up to a total of 15 coaches. Therefore I think there must be an error in the wording, since two coaches are missing.

To conclude, I encourage the authors to continue researching in this line and solving the weakness exposed in this study, in regard to previously assessing the personality and physical condition of the evaluated coaches, since they can be important variables when it comes to obtain significant differences between the results of different trainers.

Author Response

Dear Reviewer,

Thank you very much for your time and valuable comments, which all have been considered and incorporated. The detailed list of responses is given below. We hope that the modifications and explanation will be acceptable for you.

Yours sincerely,

Rydzik, corresponding author

As far as general concepts are concerned, in my opinion, this article is carried out using a correct method and contributes to the knowledge of how these parameters can vary in coaches who observe their competitors during a full day in high-level tournaments where they must overcome different eliminatory rounds. In addition, knowing the results found in the cardiovascular parameters can be very important information for those coaches at risk of suffering cardiovascular accidents.

More specifically, I think the following errors should be corrected:

  • In Table 1 there are some results with a level of significance p >0.05 that are accompanied by (ns) and others that are not. The meaning of "ns" does not appear in any legend of the table, so I deduce that its meaning is "not significant", but I think that it must be present in all the results with p > 0.05 or in none of them, since it can be deduced that p>0.05 is a non-significant result, as explained in the statistical analysis section (line 126). Similarly, as mentioned above, I think that if (ns) is included for non-significant results, it should appear in a table legend.
  • A: This has been corrected

  • On the other hand, in Table 1 the results of significance with p <0.05 are highlighted in bold, but, however, in Table 2 all the results of significance are highlighted in bold, both those with p <0, 05, like those of p>0.05.

  • A: This has been corrected
  •  
  • Finally, in line 91 it is explained that 17 coaches were investigated and yet in line 139 it states that 11 coaches observed more than one fight of their competitors and that the judokas of 4 coaches lost in their first fight and were excluded of the tournament, which adds up to a total of 15 coaches. Therefore I think there must be an error in the wording, since two coaches are missing.
  • A: This has been corrected
  •  

To conclude, I encourage the authors to continue researching in this line and solving the weakness exposed in this study, in regard to previously assessing the personality and physical condition of the evaluated coaches, since they can be important variables when it comes to obtain significant differences between the results of different trainers.

A: This has been corrected

Reviewer 3 Report

I believe the authors tap into an important topic in physiology and health.

This study aims to analyze the testosterone and cortisol response in judo coaches during a competition day. It is a hard and ambitious study.  I think some things need clarifying for the publication that will help in the overall interpretation and understanding of the results before being published within the scope of the MDPI- Biology

Introduction

Comment 1: The introduction is well written and clear. I suggest to the authors add more information about testosterone and cortisol response to competition and circadian rhythm.

Material and Methods

Comment 2: Sleep time and quality are variables that considerably influence cortisol concentrations. Authors should consider sleep variables when comparing average results of hormone concentrations.

Comment 3: As is well known, age is a key factor in endogenous testosterone production (doi:10.1371/journal.pone.0109346). The authors did not consider age in their statistical analyses. I suggest they use the normalised values for testosterone and cortisol as a function of participants' age. (doi: 10.1530/EC-17-0160).

Comment 4: Body composition is a factor that influences hormonal levels (Body Fat, Muscle Mass,....). Why the authors did not assess these variables?

Comment 5: The victory and loss, conditioning the hormone response (doi:10.1016/j.psyneuen.2012.02.011) Why the authors did not consider this variable in their analysis? All participants had the same number of victories?

Comment 6: This study has several limitations, however, the authors did not identify any limitations. Please identify the limitations and strengths of this study.

Comment 7: This study followed the Declaration of Helsinki? Please add information?

Comment 8: Please, discuss the results in the light of a new analysis.

Author Response

Dear Reviewer,

Thank you very much for your time and valuable comments, which all have been considered and incorporated. The detailed list of responses is given below. We hope that the modifications and explanation will be acceptable for you.

Yours sincerely,

Rydzik, corresponding author

I believe the authors tap into an important topic in physiology and health.

This study aims to analyze the testosterone and cortisol response in judo coaches during a competition day. It is a hard and ambitious study.  I think some things need clarifying for the publication that will help in the overall interpretation and understanding of the results before being published within the scope of the MDPI- Biology

Introduction

Comment 1: The introduction is well written and clear. I suggest to the authors add more information about testosterone and cortisol response to competition and circadian rhythm.

A: This has been corrected

Material and Methods

Comment 2: Sleep time and quality are variables that considerably influence cortisol concentrations. Authors should consider sleep variables when comparing average results of hormone concentrations.

A: This has been corrected

Comment 3: As is well known, age is a key factor in endogenous testosterone production (doi:10.1371/journal.pone.0109346). The authors did not consider age in their statistical analyses. I suggest they use the normalised values for testosterone and cortisol as a function of participants' age. (doi: 10.1530/EC-17-0160).

A: This has been corrected

Comment 4: Body composition is a factor that influences hormonal levels (Body Fat, Muscle Mass,....). Why the authors did not assess these variables?

A: Added information in limitation

Comment 5: The victory and loss, conditioning the hormone response (doi:10.1016/j.psyneuen.2012.02.011) Why the authors did not consider this variable in their analysis? All participants had the same number of victories?

A: This has been corrected

Comment 6: This study has several limitations, however, the authors did not identify any limitations. Please identify the limitations and strengths of this study.

A: This has been corrected

Comment 7: This study followed the Declaration of Helsinki? Please add information?

A: This has been corrected

Comment 8: Please, discuss the results in the light of a new analysis.

A: This has been corrected

Round 2

Reviewer 1 Report

Dear Authors

I reviewed this manuscript # biology-1685661 entitled "Stress Responses To One-Day Athletic Tournament In Sport Coaches. A Pilot Study" which you submitted to the biology. This paper appears to have been improved a lot compared to the previous version. However, there are still some areas that need to be fixed. Anyway, I would like to inform you of the consideration of your manuscript as follows.

Minor points

Line 85 should provide a reason for the necessity of conducting this study. The need and purpose of research are clearly different.

On line 88, in 'Materials and Methods' to '2.1. Participants', '2,2, Experimental Design', '2.3. Measurement methods', etc., should be presented separately.

What does 'Czas snu so liczono od momentu' in line 118 mean?

On line 134, the statistical symbol 'p' must be italicized. All 'p' values presented in the results below should be italicized in the same way.

The abbreviated variable names presented in Tables 1 and 2 are all full names and should be presented again in footnotes.

Major points

Overall, this paper is very awkward and inconsistent.

This paper is a comparison between two periods for a single group, and it is considered very difficult to generalize the results of this paper. It is common to put these restrictions at the end of the 'discussion', but you have to be very careful because there is a risk of damaging the reputation of this paper as well as the researchers in the future, and also the reputation of 'Biology'.

However, I was somewhat moved by the attempts to be evaluated on several occasions. I would like to give you another chance, so please write your thesis carefully and write your 'discussion' carefully.

Sincerely,

Author Response

Dear Reviewer,

Thank you very much for your time and valuable comments, which all have been considered and incorporated. The detailed list of responses is given below. We hope that the modifications and explanation will be acceptable for you.

Yours sincerely,

Rydzik, corresponding author

Minor points

Line 85 should provide a reason for the necessity of conducting this study. The need and purpose of research are clearly different.

A: This has been corrected

On line 88, in 'Materials and Methods' to '2.1. Participants', '2,2, Experimental

A: This has been corrected

Design', '2.3. Measurement methods', etc., should be presented separately.

A: This has been corrected

What does 'Czas snu so liczono od momentu' in line 118 mean?

A: This has been corrected

On line 134, the statistical symbol 'p' must be italicized. All 'p' values presented in the results below should be italicized in the same way.

A: This has been corrected

The abbreviated variable names presented in Tables 1 and 2 are all full names and should be presented again in footnotes.

A: This has been corrected

Major points

Overall, this paper is very awkward and inconsistent.

A: This has been corrected. The manuscript has been checked and corrected by a native speaker

This paper is a comparison between two periods for a single group, and it is considered very difficult to generalize the results of this paper. It is common to put these restrictions at the end of the 'discussion', but you have to be very careful because there is a risk of damaging the reputation of this paper as well as the researchers in the future, and also the reputation of 'Biology'.

A: Added information in the limitations of the study. Rewrote the entire text and improved the discussion.  Such a procedure was used by Hudson J et al. 2013.

However, I was somewhat moved by the attempts to be evaluated on several occasions. I would like to give you another chance, so please write your thesis carefully and write your 'discussion' carefully.

A: The manuscript has been revised by a native speaker. The discussion has been revised. 

Reviewer 3 Report

The authors did a great job and responded to my all comments.

I suggest only one minor review. Line 125 - Please update the reference to the Declaration of Helsinki to 2013.

Thank you for your attention. 

Author Response

Dear Reviewer,

Thank you very much for your time and valuable comments, which all have been considered and incorporated. The detailed list of responses is given below. We hope that the modifications and explanation will be acceptable for you.

Yours sincerely,

Rydzik, corresponding author

I suggest only one minor review. Line 125 - Please update the reference to the Declaration of Helsinki to 2013.

A: This has been corrected 

Round 3

Reviewer 1 Report

Dear Authors

I re-reviewed this manuscript # biology-1685661 entitled "Stress Responses To One-Day Athletic Tournament In Sport Coaches. A Pilot Study" which you submitted to the biology. This paper appears to have been improved a lot compared to the previous version. However, there are still some areas that need to be fixed. Anyway, I would like to inform you of the consideration of your manuscript as follows.

Minor points

Line 85 should provide a reason for the necessity of conducting this study. The need and purpose of research are clearly different. What I pointed out has been corrected.

On line 88, in 'Materials and Methods' to '2.1. Participants', '2,2, Experimental Design', '2.3. Measurement method', etc., should be presented separately. What I pointed out has been corrected.

What does 'Czas snu so liczono od momentu' in line 118 mean? What I pointed out has been corrected.

On line 134, the statistical symbol 'p' must be italicized. All 'p' values presented in the results below should be italicized in the same way. What I pointed out has been corrected.

The abbreviated variable names presented in Tables 1 and 2 are all full names and should be presented again in footnotes. It has been modified to some extent compared to before. However, a more detailed review should be made as follows.

Replace "C-cortisol, T-testosterone, T/C-changes in the T/C ratio" on line 167 to "C, cortisol; T, testosterone; T/C, changes in the T/C ratio" please.

Replace "HR- heart rate, SBP-systolic blood pressure, DBP- diastolic blood pressure" on line 173 to "HR, heart rate; SBP, systolic blood pressure; DBP, diastolic blood pressure" please.

Major points

Overall, this paper is very awkward and inconsistent.

This paper is a comparison between two periods for a single group, and it is considered very difficult to generalize the results of this paper. It is common to put these restrictions at the end of the 'discussion', but you have to be very careful because there is a risk of damaging the reputation of this paper as well as the researchers in the future, and also the reputation of 'Biology'. What I pointed out has been corrected.

However, I was somewhat moved by the attempts to be evaluated on several occasions. I would like to give you another chance, so please write your thesis carefully and write your 'discussion' carefully. What I pointed out somewhat has been corrected.

Added minor points

1) Text, as well as all tables, should be marked with a space between the number, symbol, and number.

2) In both tables, only p should be italicized, and numbers are not italicized.

3) For the p-value given on line 215, only italicize the p.

Sincerely,

Author Response

Dear Reviewer,

Thank you very much for your time and valuable comments, which all have been considered and incorporated. The detailed list of responses is given below. We hope that the modifications and explanation will be acceptable for you.

Yours sincerely,

Rydzik, corresponding author

I re-reviewed this manuscript # biology-1685661 entitled "Stress Responses To One-Day Athletic Tournament In Sport Coaches. A Pilot Study" which you submitted to the biology. This paper appears to have been improved a lot compared to the previous version. However, there are still some areas that need to be fixed. Anyway, I would like to inform you of the consideration of your manuscript as follows.

Minor points

Line 85 should provide a reason for the necessity of conducting this study. The need and purpose of research are clearly different. → What I pointed out has been corrected.

A: Thank you

On line 88, in 'Materials and Methods' to '2.1. Participants', '2,2, Experimental Design', '2.3. Measurement method', etc., should be presented separately. → What I pointed out has been corrected.

A: Thank you

What does 'Czas snu so liczono od momentu' in line 118 mean? → What I pointed out has been corrected.

A: Thank you

On line 134, the statistical symbol 'p' must be italicized. All 'p' values presented in the results below should be italicized in the same way. → What I pointed out has been corrected.

A: Thank you

The abbreviated variable names presented in Tables 1 and 2 are all full names and should be presented again in footnotes. → It has been modified to some extent compared to before. However, a more detailed review should be made as follows.

A: Thank you

Replace "C-cortisol, T-testosterone, T/C-changes in the T/C ratio" on line 167 to "C, cortisol; T, testosterone; T/C, changes in the T/C ratio" please.

A: This has been corrected 

Replace "HR- heart rate, SBP-systolic blood pressure, DBP- diastolic blood pressure" on line 173 to "HR, heart rate; SBP, systolic blood pressure; DBP, diastolic blood pressure" please.

A:This has been corrected 

Major points

Overall, this paper is very awkward and inconsistent.

This paper is a comparison between two periods for a single group, and it is considered very difficult to generalize the results of this paper. It is common to put these restrictions at the end of the 'discussion', but you have to be very careful because there is a risk of damaging the reputation of this paper as well as the researchers in the future, and also the reputation of 'Biology'. → What I pointed out has been corrected.

A: Thank you 

However, I was somewhat moved by the attempts to be evaluated on several occasions. I would like to give you another chance, so please write your thesis carefully and write your 'discussion' carefully. → What I pointed out somewhat has been corrected.

A: Thank you 

Added minor points

1) Text, as well as all tables, should be marked with a space between the number, symbol, and number.

A: This has been corrected 

2) In both tables, only p should be italicized, and numbers are not italicized.

A: This has been corrected 

3) For the p-value given on line 215, only italicize the p.

A: This has been corrected